# Walk Longer! Using Wearable Inertial Sensors to Uncover Which Gait Aspects Should Be Treated to Increase Walking Endurance in People with Multiple Sclerosis

**DOI:** 10.3390/s24227284

**Published:** 2024-11-14

**Authors:** Ilaria Carpinella, Rita Bertoni, Denise Anastasi, Rebecca Cardini, Tiziana Lencioni, Maurizio Ferrarin, Davide Cattaneo, Elisa Gervasoni

**Affiliations:** 1IRCCS Fondazione Don Carlo Gnocchi Onlus, 20148 Milan, Italy; icarpinella@dongnocchi.it (I.C.); rbertoni@dongnocchi.it (R.B.); tlencioni@dongnocchi.it (T.L.); davide.cattaneo@unimi.it (D.C.); egervasoni@dongnocchi.it (E.G.); 2Department of Biomedical Sciences, University of Sassari, 07100 Sassari, Italy; denise.anastasi89@gmail.com; 3Department of Pathophysiology and Transplantation, University of Milan, 20122 Milan, Italy; rcardini@dongnocchi.it

**Keywords:** multiple sclerosis, walking endurance, inertial sensors, dynamic balance, gait regularity, rehabilitation

## Abstract

Reduced walking endurance is common in people with multiple sclerosis (PwMS), leading to reduced social participation and increased fall risk. This highlights the importance of identifying which gait aspects should be mostly targeted by rehabilitation to maintain/increase walking endurance in this population. A total of 56 PwMS and 24 healthy subjects (HSs) executed the 6 min walk test (6 MWT), a clinical measure of walking endurance, wearing three inertial sensors (IMUs) on their shanks and lower back. Five IMU-based digital metrics descriptive of different gait domains, i.e., double support duration, trunk sway, gait regularity, symmetry, and local dynamic instability, were computed. All metrics demonstrated moderate–high ability to discriminate between HSs and PwMS (AUC: 0.79–0.91) and were able to detect differences between PwMS at minimal (PwMS_mFR_) and moderate–high fall risk (PwMS_FR_). Compared to PwMS_mFR_, PwMS_FR_ walked with a prolonged double support phase (+100%), larger trunk sway (+23%), lower stride regularity (−32%) and gait symmetry (−18%), and higher local dynamic instability (+24%). Normative cut-off values were provided for all metrics to help clinicians in detecting abnormal scores at an individual level. The five metrics, entered into a multiple linear regression model with 6 MWT distance as the dependent variable, showed that gait regularity and the three metrics most related to dynamic balance (i.e., double support duration, trunk sway, and local dynamic instability) were significant independent contributors to 6 MWT distance, while gait symmetry was not. While double support duration and local dynamic instability were independently associated with walking endurance in both PwMS_mFR_ and PwMS_FR_, gait regularity and trunk sway significantly contributed to 6 MWT distance only in PwMS_mFR_ and PwMS_FR_, respectively. Taken together, the present results allowed us to provide hints for tailored rehabilitation exercises aimed at specifically improving walking endurance in PwMS.

## 1. Introduction

Multiple sclerosis (MS) is a chronic, inflammatory, and neurodegenerative disease characterized by demyelinating lesions and axonal damage within the central nervous system [1]. With an increasing prevalence of 2.8 million people worldwide and a mean age at diagnosis of 32 years, MS represents the first cause of non-traumatic disability in young adults [2].

Although MS affects several functional domains, 70% of people with MS (PwMS) consider walking impairment as the most challenging aspect of their condition [3]. Since gait dysfunctions begin early in the disease course and gradually progress over time [4], they highly contribute to the reduction in independence and quality of life of PwMS in the most productive years of their lives [5].

Among walking impairments, the reduction in walking endurance is particularly important for PwMS [6]. Indeed, published studies found that reduced walking endurance, as measured by the six-minute walk test (6 MWT), is already present at the very early stages of MS (Expanded Disability Status Scale (EDSS) ≤ 2.5) [7], and gradually deteriorates with the progression of the disease [8,9,10]. The 6 MWT distance has found to be correlated to the number of steps performed during real-world walking [11], independence in daily life activities [12], quality of life [10], and social participation [13,14] in PwMS. Importantly, previous studies found that poor walking endurance was not only associated with past fall history [15] but also contributed to predict future falls in PwMS [16].

Given the high impact of locomotion impairments on this population, several rehabilitation approaches (including resistance, aerobic, or balance exercises, multicomponent training, exergame, robotics, yoga, and Pilates) have been proposed with good results in terms of increased 6 MWT distance [17]. However, data and analyses regarding the superiority of one intervention over another in terms of effects on walking endurance were not reported. To partly overcome this limitation, it is important to understand which specific aspects significantly contribute to gait endurance. Previous studies using clinical measures [12,18] found that self-rated fatigue was significantly associated with the 6 MWT distance. However, although perceived fatigue is among the most common symptoms of MS with a prevalence around 83% [19], it explained a percentage of the variance in the 6 MWT between 6% [12] and 30% [18], suggesting that fatigue only partially accounted for the reduction in walking endurance in PwMS. This was confirmed also by the results of Dalgas et al. [20]. Another study of Wetzel et al. [21] found that balance confidence and lower limb strength, as measured by clinical scales, were independent factors associated with the 6 MWT distance. This result was more recently confirmed by Mañago et al. [22] and Callesen et al. [23], who found that instrumentally assessed balance and strength of lower limb and trunk muscles were independent contributors to walking endurance. Taken together, these results suggested that balance and muscle strength should be targeted during rehabilitation aimed at increasing walking endurance in PwMS. In particular, resistance training, consisting of increasing muscle strength through repeated movements against a weight, in sitting or standing position, is included in the European recommendations on physiotherapy in MS [24], having demonstrated to improve muscle force in PwMS [25]. However, despite this result, the effect of resistance training on walking is controversial [25], and no specific guidelines for balance training are present in the cited recommendations. This represents a first limitation of the current physiotherapy practice for PwMS. A second limitation was acknowledged by Callesen et al. [23]: although balance and muscle strength demonstrated to significantly contribute to walking endurance in PwMS, balance was assessed only during static upright stance, and muscle force was measured in isometric condition only [23] in a sitting or lying position [22] and not during dynamic functional tasks such as walking. Taken together, these two limitations suggested that other rehabilitation paradigms and exercises should be considered as alternatives or supplements to current physiotherapy practice for gait in PwMS. Hence, to partially fill this gap, it is of paramount importance to understand which are the independent functional kinematic aspects of gait contributing most to walking endurance, in order to focus the rehabilitation exercises on them, following a potentially more efficacious, task-oriented functional approach which facilitates motor learning by promoting neuroplasticity in PwMS [26]. To the best of our knowledge, there are no published studies on this topic yet.

Based on these considerations, in the present study, wearable inertial sensors were applied on a cohort of PwMS with different levels of severity during the execution of a 6 MWT. A set of digital metrics descriptive of different kinematic aspects of gait were computed from the sensors’ signals in order (i) to analyze the ability of these metrics to detect differences between healthy subjects and PwMS, and between PwMS at minimal fall risk (PwMS_mFR_) and PwMS at moderate–high fall risk (PwMS_FR_), (ii) to provide normative cut-offs, helping clinicians to detect abnormal values at an individual level, and (iii) to analyze which metrics are independent factors significantly associated with 6 MWT distance, providing potentially useful hints to tailor rehabilitation exercises aimed at increasing gait endurance in PwMS. This third aspect was analyzed, separately, also on PwMS_mFR_ and PwMS_FR_ subgroups, as a secondary ancillary analysis.

## 2. Materials and Methods

### 2.1. Study Design

This cross-sectional study reports the results of a secondary analysis conducted on the baseline data of PwMS recruited in two previous randomized controlled trials (trial registration: www.ClinicalTrials.gov, accessed on 15 October 2024, IDs: NCT03201692 and NCT04006613). All participants, recruited at the IRCCS Fondazione Don Carlo Gnocchi (Milan, Italy), signed a written informed consent to the studies that were approved by the local Ethics Committee (codes: 11/2017/CE_FdG/SA and 7/2020/CE_FdG/FC/SA).

### 2.2. Participants

Fifty-six PwMS were recruited on the basis of the following inclusion/exclusion criteria. The inclusion criteria were age ≥ 18 years, confirmed diagnosis of MS based on McDonald criteria [27,28], no relapses in the previous two months, Expanded Disability Status Scale (EDSS) [29] score ≤ 6.5, ability to walk at least 10 m with or without an assistive device, and Mini-Mental State Examination (MMSE) [30] score ≥ 21. The exclusion criteria were an inability to understand the instructions given in the study and/or to sign the informed consent, assumption of a steroidal drug therapy during the study, presence of psychiatric complications, presence of severe joint and/or bone disorders interfering with balance and gait (based upon clinical judgment), and diagnosis of cardiovascular or other concomitant neurological diseases. A sample of 24 healthy subjects (HSs) was also recruited as a control group. The inclusion criteria for HSs were age ≥ 18 years and the absence of neurological, psychiatric, cardiovascular, and musculoskeletal diseases interfering with walking and balance.

The total sample size (56 PwMS and 24 HSs) was considered adequate based on a previous study [7] showing a 6 MWT distance (mean ± standard deviation) of 559.3 m ± 86.0 m for PwMS and of 628.0 m ± 90.7 m in HSs. These data resulted in an effect size of 0.78, indicating that 64 participants (43 PwMS and 21 HSs) were needed to obtain a difference between groups with α = 0.05, power = 0.80, and an allocation ratio of 0.5.

### 2.3. Clinical and Instrumented Assessment

PwMS were clinically assessed by experienced trained physiotherapists who administered the modified Dynamic Gait Index (mDGI) [31,32] and the six-minute walk test (6 MWT) [10]. The 6 MWT was executed also by HSs. Both assessment tools have been previously validated on PwMS and showed strong interrater and test–retest reliability [10,31].

The mDGI [31] is a clinical test assessing gait adaptability and dynamic balance during eight walking tasks under different external demands (e.g., walking with head rotation, walking over and around obstacles, stair ascending). Each item is scored from 0 to 8 points, with increasing values meaning better performances. The total score is 64 meaning normal gait adaptability and dynamic balance. Based on the cut-off values recently identified by Torchio et al. [33], PwMS were divided into individuals at minimal fall risk (PwMS_mFR_, mDGI > 49) and individuals at moderate–high fall risk (PwMS_FR_, mDGI ≤ 49).

The 6 MWT assesses walking endurance by measuring the distance (in meters) walked by the individual in 6 min. All participants executed the test following the instruction of Goldman et al. for PwMS [10]. In particular, the subjects were required to walk “as far as possible and as fast as possible” back and forth along a 30 m hallway for 6 min. The use of a walking aid was allowed if needed. All participants executed the test wearing three wireless inertial measurement units (IMUs) (MTw, Xsens, NL) attached by elastic belts on their lower trunk (L5 level) and on the lateral malleoli. Each IMU was made up of a three-dimensional accelerometer (±160 m/s^2^ range), a three-dimensional gyroscope (±1200 deg/s range), and a three-dimensional magnetometer (±1.5 Gauss range). Signals from the IMUs were acquired with a sampling rate of 75 Hz.

### 2.4. Data Processing

Data processing was performed using MATLAB R2017b (The MathWorks, Natick, MA, USA).

Trunk accelerations in the antero-posterior (AP), medio-lateral (ML), and vertical (VT) directions were reoriented to a horizontal–vertical coordinate system [34]. Afterwards, the portions of IMU signals corresponding to the 180 deg curves, at the beginning and the end of the 30 m corridor, were identified from the trunk angular velocity around the vertical axis [7,35] and excluded from the subsequent analysis. For each straight-line walking bout, instants of foot-strike and foot-off were detected from the angular velocity around the medio-lateral axis and the antero-posterior acceleration of each shank [36]. Hence, a set of 33 digital metrics (Table 1) organized in gait domains [37,38,39,40] were computed on the 10 steady-state strides in the middle of each corridor and then averaged over the total number of straight-line walking bouts executed by each participant.

The considered metrics, widely used to characterize walking in PwMS (see Woelfle et al. [41] for a recent review), have been previously validated on PwMS and other neurological pathologies (i.e., Parkinson’s disease and stroke), demonstrating moderate-to-excellent test–retest reliability (Intraclass Correlation Coefficient ≥ 0.50 [42]) in these populations [38,43,44,45,46,47,48,49].

**Table 1 sensors-24-07284-t001:** Description of the IMU-based metrics computed during the instrumented six-minute walk test.

Domain	Metric	Description
Rhythm and Pace	Stride Duration (s)	Time interval between two consecutive foot-strikes of the same leg.
	Cadence (step/min)	Reciprocal of step time, from the foot-strike of one leg to the foot-strike of the other leg.
	Stride Length (m)	Estimated as gait speed × stride duration, where gait speed is the average speed maintained during periods of straight-line walking (i.e., the corridor’ length [30 m] divided by the time needed to walk it, excluding the duration of the curves).
	Stance Duration (%)	Time interval between the foot-strike of one leg and the foot-off of the same leg, expressed as a percentage of stride duration.
	Swing Duration (%)	Time interval between the foot-off of one leg and the foot-strike of the same leg, expressed as a percentage of stride duration.
	Double Support Duration (%)	Time interval between the foot-strike of one leg and the foot-off of the contralateral leg, expressed as a percentage of stride duration.
Regularity/Variability	Stride Regularity [Modulus, AP, ML, VT] (unitless)	Amplitude of the second peak of the normalized autocorrelation function computed from the trunk acceleration modulus and from the AP, ML, and VT acceleration components. Increasing values, from 0 to 1, indicate higher stride regularity [38,50].
	Step Regularity [Modulus, AP, ML, VT] (unitless)	Amplitude of the first peak of the normalized autocorrelation function computed from the trunk acceleration modulus and from the AP, ML, and VT acceleration components. Increasing values, from 0 to 1, indicate higher step regularity [38,50].
	Stride/Step Time Variability (unitless)	Coefficient of variation (CV = standard deviation/mean) of stride/step duration [49,51]. Higher values indicate more variable stride/step time.
Gait Symmetry	Improved Harmonic Ratio—iHR [AP, ML, VT] (unitless)	Discrete fast Fourier transform was used to decompose the AP, ML, and VT acceleration of the trunk into harmonics. Hence, iHR was computed as the percentage ratio between the sum of the powers of the first ten in-phase harmonics and the sum of the powers of the first twenty harmonics (in-phase and out-of-phase) [52,53,54]. Increasing values, from 0 to 100, indicate a more symmetrical gait.
	Stride/Step Asymmetry (s)	Absolute difference between the duration of the right and left stride/step [51,55,56].
	Stance/Swing/Double Support Asymmetry (%)	Absolute difference between the duration (% of stride duration) of the right and left stance/swing/double support phases [51,55,56].
Trunk Sway	Normalized Trunk Acceleration—nRMS [AP, ML, VT] (unitless)	RMS (root mean square) value of AP, ML, and VT trunk acceleration normalized with respect to the RMS of the trunk acceleration modulus [57]. Modified from [56,58]. Higher values of this metric indicate larger trunk sway, independently from gait speed.
Gait Instability	Short-term Lyapunov Exponent Over One Stride/Step—sLyE_stride/step_ [AP, ML, VT] (unitless)	This parameter quantifies the local dynamic (in)stability of gait, reflecting the ability of the locomotor system to adapt to small perturbations naturally present during walking, such as internal control errors or external small disturbances, such as presence of obstacles or uneven surfaces [59,60,61]. sLyE was computed over one stride (sLyE_stride_) and over one step (sLyE_step_), as previously detailed [62]. In summary, trunk AP, ML, and VT accelerations pertaining to the 10 consecutive strides in the middle of each walking bout were resampled to 1000 frames (10 strides × 100 frames) [59,63] to maintain equal data length among walked corridors and participants. Afterwards, sLyE_stride_ and sLyE_step_ were calculated on the resampled AP, ML, and VT signals, following the Rosenstein method [64], with an embedding dimension (*m*) equal to 5 and a time delay (*T*) of 10 samples (*m* and *T* estimated using published algorithms [65]). Higher values of the Lyapunov exponents reflect poorer capability of the locomotor system to cope with small perturbations, thus indicating greater local dynamic instability.

AP: antero-posterior; ML: medio-lateral; VT: vertical.

### 2.5. Statistical Analysis

Statistical analysis was performed using STATISTICA 7.0 (Statsoft, Tulsa, OK, USA).

Demographics and clinical characteristics of PwMS and HSs were compared using a Mann–Whitney U test (MWt) for age and 6 MWT distance, and a chi-squared test (χ^2^) for sex distribution and number of walking aid users. Comparisons among HSs, PwMS_mFR_, and PwMS_FR_ were performed through a Kruskal–Wallis test (KWt) with the Bonferroni–Holm (BH) post hoc procedure for age and 6 MWT distance, and Fisher’s exact test (FET) for sex distribution and number of waking aid users. A Mann–Whitney U test was also used to compare time since MS diagnosis, EDSS, and mDGI score between PwMS_mFR_ and PwMS_FR_.

The Pearson’s correlation coefficient (*r*) was used to analyze the bivariate correlation between the 6 MWT distance and each of the 33 instrumented metrics reported in Table 1. This analysis was performed on PwMS data only. For each of the five gait domains (rhythm and pace, regularity/variability, gait symmetry, gait instability, and trunk sway), the metric characterized by the highest *r* value (higher bivariate association with the 6 MWT) was chosen for the subsequent analyses.

Since instrumented data were not always normally distributed, the five selected instrumented metrics were compared between PwMS and HSs using a Mann–Whitney U test, and between HSs, PwMS_mFR,_ and PwMS_FR_ using a Kruskal–Wallis test (KWt) with the Bonferroni–Holm (BH) post hoc procedure. The ability of each metric to discriminate between PwMS and HSs was assessed by computing the Area Under the Receiver Operating Characteristic (ROC) Curve (AUC). Values lower than 0.70 represent a poor discriminant ability, values between 0.70 and 0.79, moderate discriminant ability, and values greater or equal to 0.80, good discriminant ability [66]. Then, if not previously published, a normative cut-off was identified, for each digital measure, as the 95th (or the 5th) percentile of HS values depending on whether its increase (or decrease) was indicative of worse performance. Finally, the percentages of PwMS showing abnormal scores (i.e., beyond normative cut-off) were computed.

The five digital metrics were then entered, as independent variables, in a multiple linear regression model with the 6 MWT distance as the dependent variable. The aim was to analyze which gait aspects were significant independent contributors to walking endurance in PwMS. Considering that the present sample consisted of 56 PwMS, the inclusion of five independent variables was in accordance with the indication from Khamis and Kepler [67], who recommended a sample size *N* greater or equal to 20 + 5 × *M*, where *M* is the number of independent variables (in the present study, *N* = 20 + 5 × 5 = 45). The appropriateness of using multiple linear regression was assessed by analyzing whether its assumptions were met. In particular, multicollinearity was considered acceptable if the absolute value of Pearson’s correlation coefficients (*r*) between each pair of independent variables was below the threshold of 0.80, and if the variance inflation factors (VIFs) were below 10 [68]. Regarding the residuals of the regression model, their normality and homoscedasticity were assessed using, respectively, a Shapiro–Wilk test and White’s test, while their independency was assessed by checking if the Durbin–Watson value was within the 1.5–2.5 acceptable range [68]. To account for possible confounding factors, multiple regression analysis was performed also adjusting for age, sex [23], and use of walking aids.

As a secondary ancillary analysis, the above methods were applied, separately, on PwMS_mFR_ and PwMS_FR_ subgroups to assess if different aspects were associated with walking endurance in the two samples.

## 3. Results

### 3.1. Sample Description

The sample of PwMS included 6 (11%) individuals in the mild stages of the disease (EDSS: 0–2.5), 27 (48%) in the moderate stages (EDSS: 3.0–5.5), and 23 (41%) in the severe stages (EDSS: 6.0–6.5) [69].

As shown in Table 2, HSs and PwMS had comparable age and sex distribution, while PwMS showed significantly lower 6 MWT values. Twenty-four (43%) PwMS performed the 6 MWT with walking aids. In particular, 10 (18%) and 14 (25%) PwMS used, respectively, a monolateral and a bilateral assistive device.

On the basis of the cut-off value defined by Torchio et al. [33], 23 (41%) PwMS were at minimal fall risk (PwMS_mFR_, mDGI > 49), while 33 (59%) were at moderate–high fall risk (PwMS_FR_, mDGI ≤ 49). As shown in Table 2, HSs, PwMS_mFR_, and PwMS_FR_ had comparable age (p_KWt_ = 0.100) and sex distribution (p_FEt_ = 0.688), while a statistically significant difference was found in 6 MWT (p_KWt_ < 0.001) that was higher in HSs compared to PwMS_mFR_ and PwMS_FR_ (p_BH_ < 0.001) and lower in PwMS_FR_ compared to PwMS_mFR_ (p_BH_ < 0.001). PwMS_FR_ showed longer disease duration (p_MWt_ < 0.001), higher EDSS score (p_MWt_ < 0.001), lower mDGI (p_MWt_ < 0.001), and a greater number of participants using walking aids (p_χ2_ < 0.001).

### 3.2. Bivariate Correlation Analysis

The results of the bivariate correlation analysis between instrumented metrics and 6 MWT are reported in Figure 1. For each gait domain, the metric presenting the highest association with 6 MWT (purple bars in Figure 1) were the following:Double support duration for the rhythm and pace domain (*r* = −0.80);Stride regularity mod. (stride regularity computed on the modulus of trunk acceleration) for the regularity/variability domain (*r* = 0.73);iHR AP (antero-posterior improved Harmonic Ratio) for the gait symmetry domain (*r* = 0.64);nRMS ML (normalized root mean square of the medio-lateral trunk acceleration) for the trunk sway domain (*r* = −0.57);sLyE_step_ AP (antero-posterior short-term Lyapunov exponent computed over one step) for the gait instability domain (*r* = −0.44).

Only these five metrics were used in the subsequent analyses.

### 3.3. Between-Group Comparisons

Statistically significant differences were found between HSs and PwMS in all five metrics (p_MWt_ < 0.001). In particular, compared to HSs, PwMS showed longer double support phase, reduced gait regularity and symmetry, and increased dynamic instability and trunk sway (see Figure 2).

The AUC mean (95% confidence interval) reported in Figure 2 indicated moderate discriminant ability for double support and dynamic instability (AUC = 0.79), and good discriminant ability for stride regularity, gait symmetry, and trunk sway (AUC ≥ 0.80).

The results of the comparisons among HSs, PwMS_mFR_, and PwMS_FR_ (Table 3) revealed statistically significant differences among the three groups in all five instrumented metrics (p_KWt_ < 0.001). In particular, compared to HSs, both PwMS_mFR_ (p_BH_ ≤ 0.033) and PwMS_FR_ (p_BH_ < 0.001) showed alterations in all considered gait domains. These alterations were more severe in PwMS_FR_ than in PwMS_mFR_ (p_BH_ ≤ 0.046).

The normative cut-off values identified for each metric are reported in Table 4, together with the percentage of PwMS showing abnormal values. Compared to PwMS_mFR_, the PwMS_FR_ subgroup included larger percentages of individuals with abnormal digital metrics (see Table 4).

### 3.4. Multiple Linear Regression on PwMS

Table 5 shows the results of the association between walking endurance (6 MWT) and the five digital metrics (double support duration, stride regularity mod., iHR AP, nRMS ML, and sLyE_step_ AP) descriptive of the five considered gait domains on the whole sample of PwMS (*N* = 56).

The appropriateness of the analysis was confirmed, since the hypotheses of the multiple linear regression were met. In particular, multicollinearity was not a major concern since the Pearson’s correlation coefficients (*r*) between each pair of independent variables were always below the 0.80 threshold (0.23 ≤ |*r*| ≤ 0.66), and the variance inflation factors (VIFs) were always lower than 10 (VIFs between 1.2 and 2.3). The residuals of the regression model were (i) independent (Durbin–Watson value: 2.05, within the 1.5–2.5 acceptable range), (ii) normally distributed (Shapiro–Wilk test, *p* = 0.937), and (iii) homoscedastic (White’s test, *p* = 0.610).

As reported in Table 5, the model was statistically significant and explained 84% (adjusted R^2^) of the variance in the 6 MWT. Reduced double support duration (β = −0.50), increased stride regularity (β = 0.25), smaller trunk sway (β = −0.18), and lower gait instability (β = −0.21) were significantly associated (*p* ≤ 0.012) with higher walking endurance (higher 6 MWT distance). By contrast, gait symmetry was not a significant independent contributor to walking endurance (β = 0.12, *p* = 0.100). Similar results were obtained after adjusting for age, sex, and use of assistive devices (see Appendix A).

### 3.5. Secondary Analysis: Multiple Linear Regression on PwMS_mFR_ and PwMS_FR_

The results of the multiple linear regression analyses performed on the subgroups of PwMS at minimal fall risk (PwMS_mFR_, *N* = 23) and at moderate–high fall risk (PwMS_FR_, *N* = 33) are reported, respectively, in Table 6 and Table 7. In both cases, the assumptions of the analysis were met: (i) non-critical multicollinearity [VIFs lower than 10 (VIFs ≤ 2.3), and *r* lower than 0.8 (|*r*| ≤ 0.56)] and (ii) independent, normally distributed, and homoscedastic residuals [Durbin–Watson values (1.59 and 1.75) within the 1.5–2.5 range, Shapiro–Wilk test, *p* ≥ 0.215, and White test, *p* ≥ 0.49].

The model conducted on the PwMS_mFR_ subgroup (Table 6) was statistically significant and explained 74% (adjusted R^2^) of the variance in the 6 MWT. Increased walking endurance (increased 6 MWT) was significantly associated (*p* ≤ 0.049) to shortened double support duration (β = −0.58), higher stride regularity (β = 0.34), and decreased gait instability (β = −0.28). Gait symmetry and trunk sway amplitude were not significant independent contributors to walking endurance (|β| ≤ 0.25, *p* ≥ 0.103).

Regarding the PwMS_FR_ subgroup, the analysis showed that the model was statistically significant and explained 68% (adjusted R^2^) of the variance in the 6 MWT (Table 7).

Increasing walking endurance (increased 6 MWT) was significantly correlated (*p* ≤ 0.04) with shorter double support duration (β = −0.68), decreased gait instability (β = −0.24), and reduced trunk sway (β = −0.27). In contrast, stride regularity and gait symmetry were not independent contributors to 6 MWT distance (|β| ≤ 0.12, *p* ≥ 0.284) (Table 7).

Comparable results were obtained from both analyses after adjusting for age, sex, and use of assistive devices (see Appendix A).

## 4. Discussion

In the present study, a 6 MWT instrumented with three IMUs was administered to a group of HSs and PwMS to characterize, in detail, walking endurance. This is particularly relevant since the reduction in walking endurance is typical of PwMS and has a negative impact on social participation, quality of life, and fall risk. Five IMU-derived digital metrics, representative of five gait domains (rhythm and pace, regularity/variability, gait symmetry, trunk sway, and gait instability), were analyzed with the main goals of (i) assessing the ability of these metrics to discriminate between HSs and PwMS, and between PwMS at minimal (PwMS_mFR_) and at moderate–high fall risk (PwMS_FR_), (ii) providing normative cut-offs to help clinicians detect abnormal values at the individual level, and (iii) identifying the independent kinematic aspects of gait most associated with the 6 MWT that should be specifically trained during rehabilitation treatments aimed at improving walking endurance in PwMS.

### 4.1. Comparison Between Groups

As expected, walking endurance (i.e., the target variable of this study) was reduced in PwMS compared to HSs, and in PwMS_FR_ compared to PwMS_mFR_ [9], as shown by the 6 MWT distance (see Table 2).

Regarding the instrumented assessment, the five selected digital metrics (i.e., double support duration, stride regularity, antero-posterior improved Harmonic Ratio (iHR AP), antero-posterior short-term Lyapunov exponent (sLyE_step_ AP), and normalized medio-lateral trunk sway (nRMS ML)) demonstrated a moderate to high ability to discriminate between HSs and PwMS, as highlighted by the AUC values between 0.79 and 0.91. In particular, the analysis showed that, compared to HSs, PwMS were characterized by less regular and less symmetric gait (lower stride regularity and lower iHR AP), prolonged double support phase, increased antero-posterior dynamic instability (higher sLyE_step_ AP), and larger medio-lateral trunk sway (higher nRMS ML). These results are in accordance with previous studies including PwMS with different severity levels [37,38,39,40,70,71,72,73,74]. Since the selected digital metrics are known to be influenced by walking speed [63,75,76,77], these differences between HSs and PwMS could be ascribed to the lower straight-line gait velocity characterizing PwMS [median (25th; 75th percentiles): 1.0 (0.8; 1.5) m/s)] compared to HSs [1.9 (1.7; 2.2) m/s], as acknowledged also by Angelini et al. [38]. However, a subgroup analysis on ten HSs and twenty PwMS walking with comparable speed [HSs: 1.6 (1.5; 1.8) m/s, PwMS: 1.6 (1.5; 1.7), p_MWt_ = 0.202] revealed that the above results were still valid, with the exception of double support duration, which was comparable between subgroups (see Appendix A). This, in turn, suggested that the anomalies noticed in the whole sample of PwMS were not totally due to the lower velocity but could be ascribed to the pathology.

As expected, both PwMS at minimal fall risk (PwMS_mFR_) and PwMS at moderate–high fall risk (PwMS_FR_) showed significantly altered digital metrics compared to HSs. More importantly, all five parameters indicated that PwMS_FR_ walked with a more impaired gait pattern with respect to PwMS_mFR_. A previous study of Sosnoff et al. [15] on PwMS found that, compared to non-fallers, fallers had more severe disability (higher EDSS score), reduced walking endurance, and higher prevalence of walking aid use. These results were found also in the present study. In addition, Sosnoff et al. [15], Prosperini et al. [78], and Sun et al. [79] reported poorer standing balance in fallers versus non-fallers, a notion that was complemented by the present study finding higher impairments also in dynamic balance in PwMS_FR_, compared to PwMS_mFR_. In particular, with respect to PwMS_mFR_, PwMS_FR_ showed a longer double support phase, larger medio-lateral trunk sway (nRMS ML), and higher antero-posterior local dynamic instability (higher sLyE_step_ AP), therefore showing more pronounced alterations in three distinct aspects of dynamic balance.

Prolonged double support is present in PwMS from the very early stages of the disease [80,81,82], and, together with low gait speed and wide step width [83], it is part of a conservative gait strategy aimed at maintaining dynamic balance during gait by reducing the duration of the single stance phase, which implies a narrower and less stable base of support. Moreover, previous studies demonstrated that increased double support duration was associated with fear of falling [81,82,84].

Regarding trunk instability, PwMS_FR_ showed larger normalized medio-lateral trunk sway (nRMS ML) than PwMS_mFR_, suggesting higher difficulty in controlling upper body movements in the former group. To the best of our knowledge, no published studies exist using this metric to compare fallers and non-fallers in MS, so a direct comparison with the literature is not possible. Moreover, most studies reported the non-normalized RMS acceleration of the trunk, which, being strongly dependent on gait speed, resulted in an opposite finding, i.e., lower non-normalized trunk sway in PwMS compared to HSs and in more severe compared to people with less severe MS (see, for example, [38,85]). However, the normalized RMS trunk acceleration, applied to people post-stroke, found that this parameter was higher in the pathological group compared to in HSs [58], and progressively increased in persons with higher stroke severity [86]. This, in turn, suggested that this digital metric can be a good candidate to measure trunk instability, independently from walking speed.

Increased local dynamic instability of gait, quantified by the short-term Lyapunov exponent (sLyE), represents another aspect affecting dynamic balance even in PwMS with minimal disability [62,87,88]. In the present study, PwMS_FR_ showed higher local dynamic instability of overground gait compared to PwMS_mFR_, complementing previous findings related to treadmill walking [85,89].

In addition to the above results demonstrating poorer dynamic balance in PwMS_FR_ versus PwMS_mFR_, the former group was characterized also by significantly reduced stride regularity and gait symmetry than the latter. These findings were confirmed by previous studies showing that stride variability was a sound predictor of future falls in people with mild MS [90] and was strongly correlated to fall risk in people with moderate–severe MS [49]. Moreover, gait asymmetry was more marked in fallers than in non-fallers [40] and was predictive of prospective falls [91].

### 4.2. Normative Cut-Off Values

A novelty of the present study was the provision of normative cut-off values (see Table 4) for double support duration, stride regularity, antero-posterior gait symmetry (iHR AP), and medio-lateral trunk sway (nRMS ML) computed during a fast-speed 6 MWT. These cut-offs complemented the two previously reported by Carpinella et al. [7] for medio-lateral gait symmetry (iHR ML) and antero-posterior dynamic instability (sLyE_step_ AP). By applying these two thresholds, the cited study [7] found that 59% and 51% PwMS in the mild stages of the disease (EDSS: 0–2.5) showed, respectively, reduced medio-lateral gait symmetry and increased antero-posterior dynamic instability. Comparable findings were obtained in the present subsample of people with mild MS who had abnormal gait symmetry and instability in 50% of cases, thus confirming the robustness of the proposed cut-offs. Importantly, the percentages of abnormal values progressively increased in the subgroups of individuals with moderate (EDSS: 3–5.5) and severe MS (EDSS: 6–6.5) (see Appendix A), enforcing previous cross-sectional [38,39] and longitudinal [92] studies that found a worsening of digital gait metrics with increasing disability (i.e., EDDS scores). In the same vein, in the present study we found a larger percentage of abnormal values in people with more severe MS at moderate–high fall risk (PwMS_FR_) than in people with less severe MS at minimal fall risk (PwMS_mFR_) (see Table 4).

Taken together, these results are particularly relevant since the provision of normative cut-off values offers clinicians a useful tool to detect subclinical alterations from the initial stages of MS, follow their progression over time, and assess the effects of rehabilitation treatments not only in group-level analyses, but also in individual clinical judgment and decision-making.

### 4.3. Independent Constributors to Walking Endurance and Hints for Rehabilitation

A further novelty of the present study was the identification of the kinematic aspects of gait which significantly and independently contribute to walking endurance in PwMS with various severity levels (Table 5), and in PwMS at minimal (PwMS_mFR_) and moderate–high fall risk (PwMS_FR_) (Table 6 and Table 7). This, in turn, allowed us to provide hints for tailored rehabilitation exercises aimed at increasing walking endurance in this population.

The results of the multiple linear regression analysis on the whole cohort of PwMS showed that reduced double-support duration, smaller medio-lateral trunk sway (nRMS ML), lower local dynamic instability of gait (sLyE_step_ AP), and higher stride regularity were independently associated with higher 6 MWT distance, meaning increased walking endurance. While the significant contribution of double support duration and local dynamic instability was common to both PwMS_mFR_ and PwMS_FR_, stride regularity was independently associated with the 6 MWT distance in PwMS_mFR_ only, while medio-lateral trunk sway significantly contribute to walking endurance just in PwMS_FR_. This difference found in the two subgroups could be explained by previous studies on gait regularity [93] and trunk impairment [94] in PwMS. In particular, Kalron et al. [93] found that, compared to people with mild MS (EDSS: 0–3.5), an abrupt increase of two- to three-fold in gait variability emerged in people with moderate MS with an EDSS between 4 and 5.5, which represents the disability range of nearly 50% of our sample of PwMS_mFR_. In contrast, compared to people with moderate MS, a less steep increment of gait variability was found inpeople with severe MS (EDSS: 6–6.5), representing 67% of the PwMS_FR_ subgroup. These findings can therefore explain why gait regularity was independently associated with walking endurance only in PwMS_mFR_. As for the second difference, i.e., the role of the trunk, Verheyden et al. [94] found that clinically measured trunk impairment was strongly correlated with disease severity in a group of PwMS with mild to very severe walking impairment. We can therefore speculate that increased trunk sway impacts walking function mostly in people with severe MS, who represent the majority of our sample of individuals at moderate–high fall risk (PwMS_FR_).

Overall, the above results indicated that all three aspects of dynamic balance here considered (i.e., double support duration, trunk motion, and local dynamic instability of gait) should be specifically trained during walking endurance rehabilitation, with trunk sway deserves particular attention in PwMS_FR_ with a more severe disease level.

Regarding the first aspect of balance, i.e., double support duration, exercises promoting its reduction could be focused on prolonging the single support phase, which is of paramount importance not only for level walking, but also for other daily life activities including dressing and stairway walking [95]. Possible tasks to improve the single support phase could include both static (i.e., one-leg stance) and dynamic (i.e., stair ascent/descent) exercises. Standing on one leg requires both the correct body weight shift toward the stance leg and the maintenance of unipodal balance by controlling the vertical alignment and the sway of different body segments [95]. These two aspects are both impaired in PwMS [96,97,98], and their training can be facilitated by using technological devices that provide subjects with real-time visual, auditory [99], or vibrotactile feedback [100] about their performance, increasing motor learning [101]. The difficulty of the one-leg stance task could be increased, according to the severity of the disease, by adding a concurrent cognitive task (dual-task paradigm) and/or by changing the sensory conditions (i.e., standing eyes open/closed on a rigid/foam surface [97] or moving eyes/head to look at a stationary or moving target [102]). This would challenge the integration of visual, proprioceptive, and vestibular information, which is altered in PwMS [103]. In case of people with severe MS at moderate–high fall risk (PwMS_FR_), this task should be recommended to be performed in parallel bars, or near a handrail or assistive device, encouraging the person to try to use these aids with a light touch only. Single-leg stance can also be improved by increasing the strength of the muscles mainly involved in this task (e.g., hip abductors, knee extensors/flexors, ankle dorsiflexors/plantar flexors [104]) through resistance training [102,105] or by following a more functional approach in which the muscles are trained during a dynamic daily life task. In particular, stairway walking, requiring the loading of the entire body weight on one leg at a time to move the body upward/downward and forward to the next step, seems a good candidate to strengthen, in a more ecological way, the muscles involved in monopodal stance, particularly the proximal ones [106]. Importantly, considering that the single support phase of one leg corresponds to the swing phase of the contralateral leg, performing stair negotiation, in particular stair ascending, would allow also to train the hip, knee, and ankle joint muscles, working in synergy to control leg swing and avoid contact with the intermediate step. This is of paramount importance since similar muscles are also involved in the swing phase of level walking to control limb progression and potentially contribute to increase walking endurance [107].

Exercising by stairway walking could also play a role in improving the second aspect of dynamic balance here considered, i.e., trunk sway. In particular, in healthy individuals, stair descent was found to challenge dynamic balance more than stair ascent due to greater and faster trunk sway in the frontal plane [108]. To our knowledge, no published studies exist comparing stair ascent and descent in PwMS. However, a study by Carpinella et al. [109] found that during stair ascent, trunk sway in the sagittal plane was significantly larger compared to in healthy controls and people with other neurological disorders. Since trunk sway was an independent contributor to walking endurance in the whole sample of PwMS and in PwMS_FR_ but not in PwMS_mFR_, more attention should be devoted to this aspect when treating people with more severe MS at moderate–high fall risk. In these cases, the clinician should encourage the participant to ascend and descend one or two steps only, using the handrail if necessary. Regarding trunk movements during stair negotiation, again, biofeedback about the amplitude or velocity of trunk sway [100] could help to reduce trunk instability, together with core and pelvis stability exercises aimed at improving upper body muscle strength [102,110] in people with more severe MS also.

The third aspect of balance independently associated with 6 MWT in both PwMS_mFR_ and PwMS_FR_ was local dynamic instability of gait. This suggested that, to increase walking endurance, PwMS should practice walking in several contexts to improve the ability of the sensorimotor system to adapt dynamic balance to different external environmental demands typical of daily living. Indeed, local dynamic instability quantifies the capability of the locomotor system to manage small disturbances commonly occurring during gait, for example, when walking in challenging conditions [111]. Previous studies on healthy adults showed that, compared to straight-line unperturbed walking, local dynamic instability significantly increased during turning [112]. This is the same when walking on a slightly moving surface [113,114] or in the presence of visual field perturbations [114], walking while texting on a phone [115], changing gait speed [116], or walking on inclined or uneven surfaces [60,61]. Regarding PwMS, a recent study found that rotating the head while walking increased local dynamic instability compared to healthy controls, even in PwMS showing normal straight-line locomotion [57]. In addition, a study of Craig et al. [117] found that, compared to healthy subjects and non-faller PwMS, faller PwMS were characterized by higher local dynamic instability when visual information was altered by wearing glasses with prism film [117]. Taken together, these results suggested that some of the above challenging tasks, tailored on each patient’s functional status, could be included in the rehabilitation programs aimed at improving dynamic stability and increasing walking endurance in PwMS.

Finally, the present results showed that stride regularity was also independently correlated to the 6 MWT distance in the whole group of PwMS and in PwMS_mFR_, but not in PwMS_FR_. This is in line with previous studies involving people with mild–moderate MS without fall history [118,119], which found a significant association between increased gait variability and higher energy cost of walking, the latter contributing to reduce walking endurance [120]. Decreased gait regularity in PwMS has been ascribed to several aspects [72], including footfall placement variability and reduced ability to walk straight by drifting from side to side [121]. As proposed by Jones and van Emmerik [122], somatosensory deficit of the plantar surface, together with proprioceptive [123] and vestibular impairments [124] typical of MS, may play a major role in incorrectly placing the feet during locomotion and maintaining straight-line walking, thus decreasing stride regularity and dynamic balance. Regarding this point, dynamic exercises performed in different perceptive contexts should be included to challenge the proprioceptive and vestibular sensory systems, for example, walking with turns, and walking backward and sideways with eyes open and closed on rigid or compliant surfaces [102], or wearing insoles of different rigidity/compliance. A promising option to improve gait quality in general and particularly to reduce gait variability could be the use of technological devices, such as treadmills combined with virtual reality [125], which could help users to adjust their foot placement and to improve the rhythmicity and regularity of their locomotion by visualizing, directly on the mat, target objects to be reached with their feet.

In contrast to the above gait quality aspects, gait symmetry, as measured by iHR AP, was not a significant independent contributor to walking endurance, despite its strong reduction in PwMS compared to HSs and its high bivariate correlation with the 6 MWT distance (*r* = 0.64, see Figure 1). A possible explanation could be that its contribution was “masked” by that of the other considered aspects. In particular, the iHR AP metric showed a strong correlation with stride regularity (*r* = 0.66). Although this Pearson’s *r* value is below the threshold for acceptable multicollinearity between independent variables (i.e., 0.80 [68]), it suggests that the two digital metrics could measure partially dependent constructs. In this vein, the previous literature found that gait asymmetry reduced walking rhythmicity and regularity in older adults [126]. From this point of view, the biofeedback rehabilitation exercises described above and aimed at increasing gait regularity can simultaneously improve step symmetry also. Another explanation is that, in a population of people with predominantly moderate–severe MS, dynamic balance and gait regularity are the main aspects to focus on during rehabilitation training aimed at increasing walking endurance. Future studies on people with less severe MS should be conducted to corroborate these findings and design tailored preventive rehabilitation treatments to slow gait deterioration [127].

### 4.4. Study Limitations

Some limitations must be acknowledged regarding the present study. First, the cross-sectional design of the present study does not allow us to claim that there is a cause–effect relationship between walking endurance and the digital metrics selected to describe gait quality. Second, we did not consider the effect of fatigue, given previous results reporting only a small contribution of self-reported fatigue to the reduction in walking endurance in PwMS [12,18]. However, future studies should be performed to analyze objective performance fatigability, i.e., the change in gait quality indexes during prolonged walking. Third, the cohort of HSs tested was small. Larger samples of healthy individuals with different ages should be analyzed to provide more robust normative cut-off values for the instrumented parameters. Fourth, the present group of PwMS included only six (11%) persons in the mild stages of the disease. Future studies should be devoted to early-stage populations to complement present findings. Fifth, the results of the secondary ancillary multiple regression analyses on PwMS_mFR_ and PwMS_FR_ should be interpreted with caution given the small sizes of the two subgroups. Finally, the 6 MWT was instrumented with three inertial sensors requiring a laptop to record the kinematic signals analyzed. Other technologies that are even more user-friendly could be used to compute the same data via smartphones [128,129] or other variables such as the displacements of the center of pressure during functional tasks though pressure insoles [130].

## 5. Conclusions

The five digital metrics computed from an instrumented 6 MWT, i.e., double support duration, stride regularity, iHR AP, nRMS ML, and sLyE_step_ AP, demonstrated moderate–high ability to discriminate between HSs and PwMS (AUC: 0.79–0.91) and were able to detect differences between PwMS at minimal (PwMS_mFR_) and moderate–high fall risk (PwMS_FR_). In particular, compared to PwMS_mFR_, PwMS_FR_ walked with a prolonged double support phase, larger medio-lateral trunk sway (higher nRMS ML), lower stride regularity, reduced gait symmetry (lower iHR AP), and higher antero-posterior local dynamic instability (higher sLyE_step_ AP). Normative cut-off values were provided for these metrics to help clinicians in detecting abnormal scores at an individual level. The results of the multiple regression analyses showed that stride regularity and the three metrics most related to dynamic balance (i.e., double support duration, trunk sway, and local dynamic instability) were significant independent contributors to 6 MWT distance, while gait symmetry was not, although it was the aspect which most discriminated between HSs and PwMS and between PwMS subgroups. While double support duration and local dynamic instability were independently associated with walking endurance in both PwMS_mFR_ and PwMS_FR_, stride regularity and trunk sway were independent contributors to the 6 MWT only in PwMS_mFR_ and PwMS_FR_, respectively. Taken together, the present results allowed us to provide hints for tailored rehabilitation exercises aimed at specifically improving walking endurance in PwMS.

## Figures and Tables

**Figure 1 sensors-24-07284-f001:**
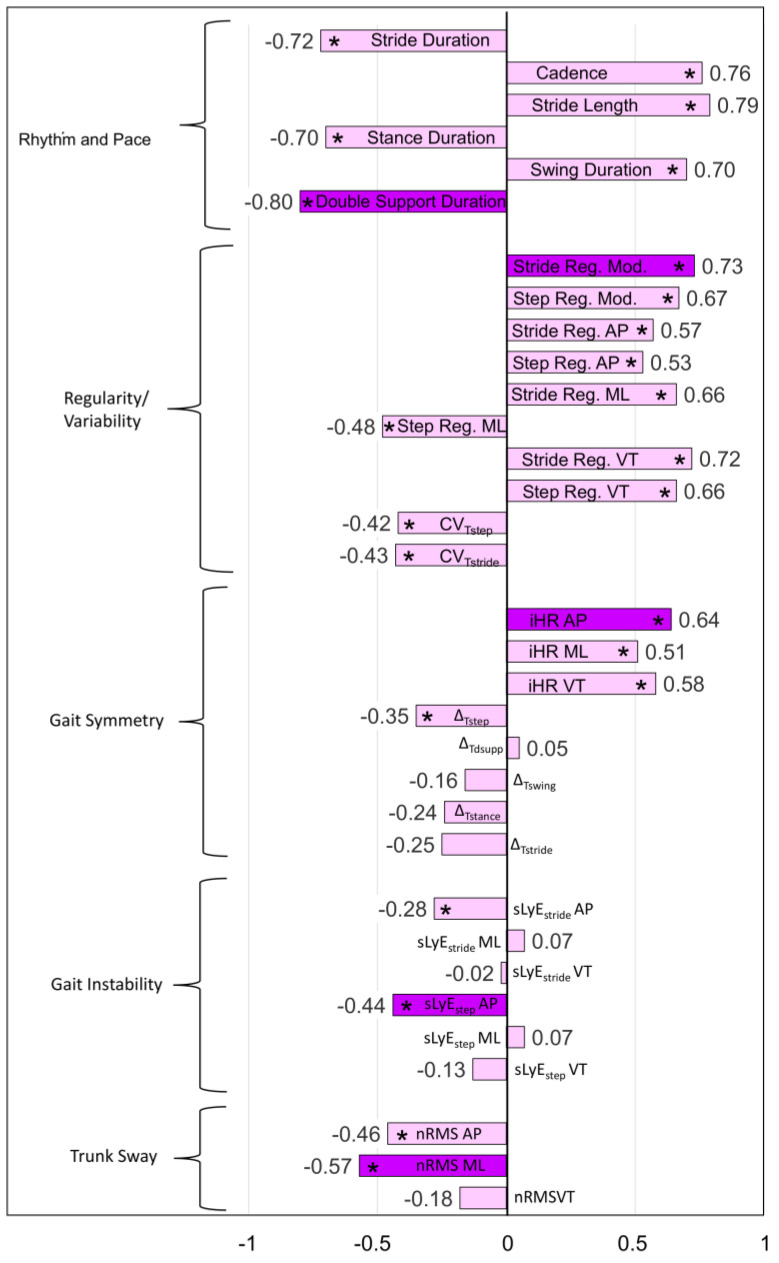
Pearson’s correlation coefficient *r* between six-minute walk test distance and IMU-based digital metrics descriptive of the gait domains reported on the left. The metric showing the highest correlation for each domain is reported in dark violet. * *p* < 0.05. Reg.: regularity; CV: coefficient of variation; iHR: improved Harmonic Ratio; ∆: absolute difference between right and left side; T_step_: step duration; T_stride_: stride duration; T_stance_: stance duration; T_swing_: swing duration; T_dsupp_: double support duration; nRMS: normalized root mean square of trunk acceleration; sLyE_stride/step_: short-term Lyapunov exponent computed over one stride/step; Mod.: trunk acceleration modulus; AP: antero-posterior; ML: medio-lateral; VT: vertical.

**Figure 2 sensors-24-07284-f002:**
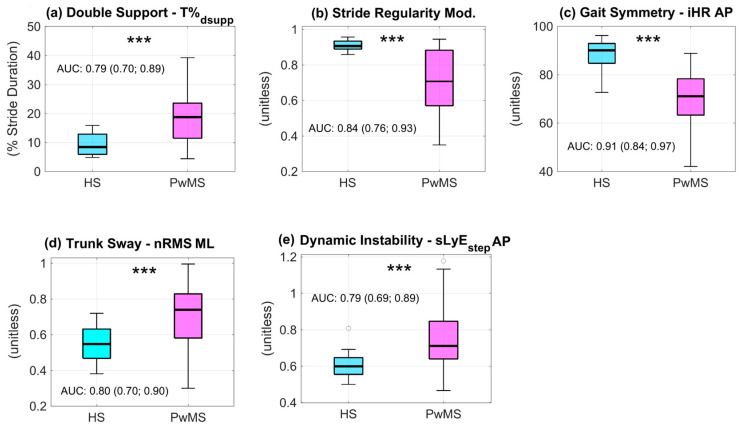
Digital metrics descriptive of gait in healthy subjects (HSs) and people with MS (PwMS). Bold line: median; Box: interquartile range; Whisker: range. *** *p*< 0.001 (HSs vs. PwMS, Mann–Whitney U Test). iHR: improved Harmonic Ratio; nRMS: normalized root mean square of trunk acceleration; sLyE_step_: short-term Lyapunov exponent computed over one step; Mod.: trunk acceleration modulus; AP: antero-posterior; ML: medio-lateral; AUC: Area Under the Receiver Operating Characteristic (ROC) Curve, mean (95% confidence interval).

**Table 2 sensors-24-07284-t002:** Demographic and clinical characteristics of healthy subjects and people with MS.

	HSs (*N* = 24)	PwMS (*N* = 56)	*p*-Value	PwMS_mFR_ (*N* = 23)	PwMS_FR_ (*N* = 33)
Age (years)	48 (41; 61)	52 (46; 63)	0.311	50 (40; 57)	54 (46; 66)
Female (number, %)	13 (54%)	30 (54%)	0.961	14 (61%)	16 (48%)
Time Since Diagnosis (years)	-	16 (7; 26)	-	7 (5; 19.0)	22 (12; 30)
EDSS (0–10)	-	5.0 (4.0; 6.0)	-	4.0 (2.5; 4.5)	6.0 (5.5; 6.5)
mDGI (0–64)	-	43 (35; 57)	-	58 (55; 63)	35 (31; 39)
6 MWT (m)	614 (532; 660)	348 (251; 450)	<0.001	484 (426; 513)	268 (221; 316)
Walking Aid Users (number, %)	0 (0%)	24 (43%)	<0.001	1 (4%)	23 (70%)

Values are median (25th; 75th percentiles) or number (percentage). HSs: healthy subjects; PwMS: people with MS; PwMS_mFR_: people with MS at minimal fall risk; PwMS_FR_: people with MS at moderate–high fall risk; EDSS: Expanded Disability Status Scale; mDGI: modified Dynamic Gait Index; 6 MWT: six-minute walk test. *p*-Value: result of the comparison between HSs and PwMS (chi-squared test for sex distribution and walking aid users; Mann–Whitney U test for age and 6 MWT).

**Table 3 sensors-24-07284-t003:** Comparison among HSs, PwMS_mFR_, and PwMS_FR_.

Domain Metric	HSs (*N* = 24)	PwMS_mFR_ (*N* = 23)	PwMS_FR_ (*N* = 33)	*p*-Value
**Rhythm and Pace** Double Support [%stride duration]	8.4 (5.9; 12.9)	11.5 * (8.9; 15.4)	23.0 *^†^ (18.7; 31.3)	<0.001
**Regularity/Variability** Stride Regularity Mod. [unitless]	0.91 (0.89; 0.93)	0.89 * (0.80; 0.92)	0.6 1 *^†^ (0.48; 0.69)	<0.001
**Gait Symmetry** iHR AP [unitless]	90.1 (84.7; 92.9)	78.9 * (76.4; 85.9)	65.0 *^†^ (62.7; 69.9)	<0.001
**Trunk Sway** nRMS ML [unitless]	0.55 (0.47; 0.63)	0.65 * (0.50; 0.85)	0.80 *^†^ (0.67; 0.94)	<0.001
**Gait Instability** sLyE_step_ AP [unitless]	0.60 (0.55; 0.65)	0.66 * (0.60; 0.74)	0.82 *^†^ (0.65; 0.90)	<0.001

Values are median (25th; 75th percentiles). HSs: healthy subjects; PwMS_mFR_: people with MS at minimal fall risk; PwMS_FR_: people with MS at moderate–high fall risk; iHR: improved Harmonic Ratio; sLyE_step_: short-term Lyapunov exponent over one step; nRMS: normalized root mean square of trunk acceleration; Mod.: trunk acceleration modulus; AP: antero-posterior; ML: medio-lateral. *p*-Value: result of the Kruskal–Wallis test (HSs vs. PwMS_mFR_ vs. PwMS_FR_). Symbols ‘*’ and ‘^†^’ indicate, respectively, a statistically significant difference with respect to HSs and PwMS_mFR_ (Bonferroni–Holm post hoc test).

**Table 4 sensors-24-07284-t004:** Number (percentage) of PwMS showing abnormal values of instrumented metrics.

Domain Metric	Cut-Off Value	PwMS (*N* = 56)	PwMS_mFR_ (*N* = 23)	PwMS_FR_ (*N* = 33)
**Rhythm and Pace** Double Support	>15.8 %stride duration	32 (57%)	5 (22%)	27 (82%)
**Regularity/Variability** Stride Regularity Mod.	<0.87 unitless	28 (50%)	3 (13%)	25 (76%)
**Gait Symmetry** iHR AP	<80.2 unitless	45 (80%)	12 (52%)	33 (100%)
**Trunk Sway** nRMS ML	>0.70 unitless	32 (57%)	10 (44%)	22 (67%)
**Gait Instability** sLyE_step_ AP	>0.67 unitless	33 (59%)	11 (48%)	22 (67%)

Cut-off values were set equal to the 5th percentile of healthy subject data for stride regularity and iHR AP, and to the 95th percentile for double support and nRMS ML. Cut-off for sLyE_step_ AP was taken from the literature [7]. PwMS_mFR_: people with MS at minimal fall risk; PwMS_FR_: people with MS at moderate–high fall risk; iHR: improved Harmonic Ratio; sLyE_step_: short-term Lyapunov exponent over one step; nRMS: normalized root mean square of trunk acceleration; Mod.: trunk acceleration modulus; AP: antero-posterior; ML: medio-lateral.

**Table 5 sensors-24-07284-t005:** Results of the multivariate linear regression analysis with the six-minute walk test (6 MWT) as the dependent variable (whole sample of PwMS, *N* = 56).

Adjusted R^2^	*p*-Value (F_5,50_)	Independent Variable (Domain)	b (SE)	β (SE)	*p*-Value (t_50_)
0.84	<0.001 (59.11)	Double Support * (Rhythm and Pace)	−5.45 (0.71)	−0.50 (0.06)	<0.001 (−7.69)
		Stride Regularity Mod. * (Regularity/Variability)	178.65 (57.94)	0.25 (0.08)	0.003 (3.08)
		iHR AP (Gait Symmetry)	1.47 (0.88)	0.12 (0.07)	0.100 (1.68)
		nRMS ML * (Trunk Sway)	−116.67 (44.62)	−0.18 (0.07)	0.012 (−2.61)
		sLyE_step_ AP * (Gait Instability)	−166.81 (46.95)	−0.21 (0.06)	<0.001 (−3.55)

PwMS: people with MS; iHR: improved Harmonic Ratio; sLyE_step_: short-term Lyapunov exponent over one step; nRMS: normalized root mean square of trunk acceleration; Mod.: trunk acceleration modulus; AP: antero-posterior; ML: medio-lateral; SE: standard error. * *p*-Value < 0.05.

**Table 6 sensors-24-07284-t006:** Results of the multivariate linear regression analysis with the six-minute walk test (6 MWT) as the dependent variable (subsample of PwMS_mFR_, *N = 23*).

Adjusted R^2^	*p*-Value (F_5,17_)	Independent Variable (Domain)	b (SE)	β (SE)	*p*-Value (t_17_)
0.74	<0.001 (13.34)	Double Support * (Rhythm and Pace)	−7.76 (1.88)	−0.58 (0.14)	<0.001 (−4.12)
		Stride Regularity Mod. * (Regularity/Variability)	250.98 (114.44)	0.34 (0.16)	0.042 (2.19)
		iHR AP (Gait Symmetry)	−2.24 (1.30)	−0.24 (0.14)	0.103 (−1.72)
		nRMS ML (Trunk Sway)	−101.68 (60.43)	−0.25 (0.16)	0.111 (−1.68)
		sLyE_step_ AP * (Gait Instability)	−185.08 (87.48)	−0.28 (0.13)	0.049 (−2.12)

PwMS_mFR_: people with MS at minimal fall risk; iHR: improved Harmonic Ratio; sLyE_step_: short-term Lyapunov exponent over one step; nRMS: normalized root mean square of trunk acceleration; Mod.: trunk acceleration modulus; AP: antero-posterior; ML: medio-lateral; SE: standard error. * *p*-Value < 0.05.

**Table 7 sensors-24-07284-t007:** Results of the multivariate linear regression analysis with the six-minute walk test (6 MWT) as the dependent variable (subsample of PwMS_FR_, *N = 33*).

Adjusted R^2^	*p*-Value (F_5,27_)	Independent Variable (Domain)	b (SE)	β (SE)	*p*-Value (t_27_)
0.68	<0.001 (14.83)	Double Support * (Rhythm and Pace)	−4.40 (0.67)	−0.68 (0.10)	<0.001 (−6.55)
		Stride Regularity Mod. (Regularity/Variability)	65.72 (60.24)	0.12 (0.11)	0.284 (1.09)
		iHR AP (Gait Symmetry)	0.61 (1.01)	0.06 (0.11)	0.550 (0.60)
		nRMS ML * (Trunk Sway)	−113.72 (52.76)	−0.27 (0.11)	0.040 (−2.16)
		sLyE_step_ AP * (Gait Instability)	−109.94 (46.87)	−0.24 (0.10)	0.027 (−2.35)

PwMS_FR_: people with MS at moderate–high fall risk; iHR: improved Harmonic Ratio; sLyE_step_: short-term Lyapunov exponent over one step; nRMS: normalized root mean square of trunk acceleration; Mod.: trunk acceleration modulus; AP: antero-posterior; ML: medio-lateral; SE: standard error. * *p*-Value < 0.05.

## Data Availability

The dataset used and/or analyzed during the current study is available from the corresponding author upon reasonable request.

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
