# Peer review of "Walk Longer! Using Wearable Inertial Sensors to Uncover Which Gait Aspects Should Be Treated to Increase Walking Endurance in People with Multiple Sclerosis"

_sensors, 2024, doi:10.3390/s24227284_

Round 1

Reviewer 1 Report

Comments and Suggestions for Authors

This study was conducted on patients with Multiple Sclerosis and examines their way of walking (gait) and risk of falling. Specifically, the study aims to identify which gait aspects should be targeted during rehabilitation in order to improve the patients’ walking endurance. The patients were subjected to a 6-minute walk test, which evaluates walking endurance by measuring the distance patients can walk in 6 minutes. They wore 3 inertial sensors (IMUs) on their shanks and lower back, from which 5 different gait aspects were extracted (gait regularity, gait symmetry, double support duration, trunk sway, and local dynamic instability). Each aspect contains multiple metrics/parameters. All gait aspects were able to discriminate between patients and healthy subjects, as well as between patients with high and minimal risk of falling. A minimal regression model then showed that all gait aspects except gait symmetry were independently and significantly contributing to the total distance that subjects were able to walk within 6 minutes.

I really liked reading this paper, and I think it is a very good piece of research. In my opinion the paper is very well structured and written. More importantly, the paper addresses a very important need, as medical rehabilitation needs to be targeted in order to be effective. It then becomes important to uncover these underlying correlations between various parameters in order to understand how to tailor the structure of a rehabilitation program designed to improve the walking endurance of patients with Multiple Sclerosis. The literature review section of the introduction references appropriate literature and the list of references is extensive. The motivation behind the study is presented clearly. The novelties contained in the study are also clearly highlighted. The sample size is extensive (56 patients and 24 healthy subjects). The results are supported by the data contained in the manuscript. I recommend the acceptance of this paper and only have a few comments.

Comments:

·         I understand the statistical results but is there an intuitive explanation for why gait regularity and trunk sway were only independently contributing to walking endurance in patients with minimal and moderate-high risk of falling, respectively?

·         Is there a variation of the correlations with respect to the MS severity of each patient? For example, are the correlations stronger in patients with severe MS, or maybe at an early rehabilitation stage? I understand that it is difficult to recruit patients with exactly the same severity of MS and rehabilitation stage.

Reviewer 2 Report

Comments and Suggestions for Authors

INTRODUCTION

While the introduction identifies a gap in understanding the independent functional kinematic aspects of gait contributing to walking endurance, it could more explicitly state how this gap impacts current rehabilitation practices. A clearer articulation of the consequences of this gap would strengthen the argument for the study's relevance.

METHODS

While the modified Dynamic Gait Index (mDGI) and the six-minute walk test (6MWT) are described, additional information about the qualifications of the examiners or inter-rater reliability for these assessments is lacking. This information is crucial for assessing the validity and reliability of the clinical evaluations.

RESULTS

This is a very well written and structured results section with informative and high quality figures and tables.

DISCUSSION

The discussion should b supported by a few sentences about other wearable technologies (as this seems like a significant aspect of the study) that can be used to collect gait biomechanical metrics such as smarthphones (https://journals.sagepub.com/doi/10.1177/20552076241257054), wireless pressure insoles (https://www.nature.com/articles/s41598-023-41622-3) and perhaps more practical IMUs than the one tested here. Reader would certainly benefit from understanding other technologies that can be used for this purpose.

Round 2

Reviewer 2 Report

Comments and Suggestions for Authors

Authors fully addressed my comments, thank you.